# Improved Bioherbicidal Efficacy of *Bipolaris eleusines* through Herbicide Addition on Weed Control in Paddy Rice

**DOI:** 10.3390/plants11192659

**Published:** 2022-10-10

**Authors:** Jianping Zhang, Guifang Duan, Shuang Yang, Liuqing Yu, Yongliang Lu, Wei Tang, Yongjie Yang

**Affiliations:** 1State Key Laboratory of Rice Biology, China National Rice Research Institute, Hangzhou 310006, China; 2Shanghai Greentech Laboratory Co., Ltd., Shanghai 201612, China

**Keywords:** conidial agent, biological weed control, herbicides, synergy, barnyardgrass, rice

## Abstract

*Bipolaris eleusines* was mixed with herbicides to improve the control of barnyardgrass (*Echinochloa crus-galli*), a noxious weed in rice fields. The compatibility of *B. eleusines* with herbicides was evaluated for toxic effects on spore germination and mycelium growth in vitro tests, and varied effects were observed with different chemical products. Briefly, 25 g/L penoxsulam OD plus 10% bensulfuron-methyl WP were much more compatible with *B. eleusines*, and there was no inhibition of spore germination but the promotion of mycelium growth of *B. eleusines* at all treatment rates. Under greenhouse conditions, the coefficient of the specificity of *B. eleusines* conidial agent was determined as 3.91, closer to the herbicidal control of 2.89, showing it is highly specific between rice and barnyardgrass. Field experiments in 2011 and 2012 showed that *B. eleusines* conidial agent displayed good activity on barnyardgrass, monochoria [*Monochoria vaginalis* (Burm.f.) Presl. Ex Kunth.], and small-flower umbrella sedge (*Cyperus difformis* L.) and had no negative impact on the rice plant. It also reduced the loss of rice yield when compared with the non-treated control and could make this pathogen a conidial agent for commercial bioherbicidal development in the future.

## 1. Introduction

Barnyardgrass (*Echinochloa crus-galli*) is one of the most noxious weed species around the world [1,2,3]. It may cause severe losses in both yield and quality in rice production [4,5]. Chemical herbicides are generally the mainstay for weed control practices in many countries and are responsible for much of the unparalleled increased crop productivity [6,7]. However, the repeated, intensive, and indiscriminate use of the same herbicides has brought about ecological problems such as weed resistance [8,9,10], agro-ecosystem contamination, and deterioration [11,12]. In addition, the high costs involved in developing and registering chemical herbicides have prompted researchers to investigate alternative systems to control barnyardgrass and alleviate environmental and ecological concerns [6].

Microbial bioherbicide, with a different mode of action on weeds from that of chemical herbicide, is an emerging weed control strategy toward sustainable agriculture [13] and has the potential to replace or substitute some of the current chemical herbicides for the effective control of barnyardgrass [14]. To date, the most biologically effective alternatives to chemical weed control agents that have been extensively evaluated are plant pathogens, more specifically, plant pathogenic fungi such as mycoherbicides [6]. However, mycoherbicides are typically weed species-specific which causes it to be difficult to control multiple weeds at one time [15], and some pathogenic fungi spore types of products have suffered from poor efficacy or control in the field. One possible solution to the problem was the addition of surfactants and other adjuvants into a formulation [16]. Another one was to combine the spore biocontrol agent with chemical herbicide. Therefore, compatibility with herbicides used to control weeds is often studied [17,18,19,20,21]. There were synergistic responses to sub-lethal doses of 2,4-D (2,4-dichlorophenoxyaceticacid) followed 4, 8, or 16 d later by *Phoma herbarum* inoculation under growth room and field conditions [22]. Combining *Colletotrichum truncatum* with MCPA (2-methyl-4-chlorophenoxyacetic acid), 2,4-D ester, clopyralid, or metribuzin at a 1× rate also resulted in synergistic or additive interaction [23], and 7 × 10^6^ spores/ml of *C. truncatum* plus metribuzin killed the majority of older chamomile plants, whereas the herbicide alone did not cause plant mortality [24]. All this research showed the integration of chemical and biological agents had the potential for enhanced weed control.

*Bipolaris eleusines* is a species of the genus *Bipolaris*, belonging to the family Pleosporaceae [25], a typical plant-pathogenic fungus infecting plant leaves and stems. Previous studies reported several bioactive natural products sesquiterpenoids isolated from *B. eleusines* showed anti-cancer, anti-bacterial, and anti-fungal activities [26,27,28,29]. In this study, *B**. eleusines* obtained from naturally infected barnyardgrass was evaluated as a potential biological control agent for barnyardgrass. Under greenhouse conditions, the severity of the disease index increased with increasing inoculum concentration from 1 × 10^5^ to 1 × 10^7^ spore/mL but dropped the incidence of disease in the rice field [30]. The objectives of this study were to: (1) screen compatible chemical herbicides with *B. eleusines*; (2) improve the biological efficacy of the *B. eleusines* conidial agent combined with herbicides in the greenhouse; (3) verify weed control activity and impact on rice in the field. It was critical to deciding on the development of this agent as a large-scale commercial application in the future.

## 2. Results

### 2.1. Effects on B. eleusines Viability and Growth with Herbicide Mixtures

There were different effects of five chemical herbicides, popularly used in China, on the spore germination and mycelium growth of *B. eleusines* in Table 1.

And 10% cyhalofop-butyl EC, or a combination of 25 g/L penoxsulam OD plus 10% bensulfuron-methyl or pyrazosulfuron-ethyl WP, were much more compatible with *B. eleusines* and there was no inhibition of the spore germination of *B. eleusines* at all. However, 50% quinclorac WP, 10% bensulfuron-methyl WP, and 10% pyrazosulfuron-ethyl WP alone were noticeably more inhibitive, showing 50.6–61.3% inhibition after 12 h exposure at the recommended rates or half of the total.

Relatively, the addition of 10% bensulfuron-methyl, 10% pyrazosulfuron-ethyl WP, or 25 g/L penoxsulam OD plus 10% bensulfuron-methyl WP could promote the radial growth of *B. eleusines* mycelium at nearly all test rates (Table 1). Moreover, it is a dose-dependent increase both in the survival rate and the growth rate of *B. eleusines* in response to these herbicides, suggesting that these treatments were not significantly inhibitory at the high rates likely to be encountered in a foliar application. Other herbicide treatments caused no influence or significant inhibition to mycelium growth at the recommended rates or half rates.

### 2.2. Weed Control of B. eleusines Combined Herbicides in Greenhouse

Table 2 and Table 3 presented the weed control efficacy and crop safety of *B. eleusines* conidial control agent on barnyardgrass and three rice varieties after 4 weeks of treatment.

Rice symptoms of fungal infection were regularly observed 28 d after treatment. There was no visible damage to rice growth found. As shown in Table 2, the conidial agent of *B. eleusines* was safe at 28 DAT on three rice varieties, including japonica Ribenqing, indica 9311, and glutinous rice Guixiangsinuo P106. The fresh weight inhibition rate fluctuated between −5.35~0.38%, −6.02~−2.26%, and −5.15~0.24%, respectively, but all were lower than 10%. There was no significant difference compared to herbicide control. It indicated that their ED_10_ was higher than 360 g a. i. ha^−1^.

As shown in Table 3, the inhibition rate of the fresh weight of 3.75~180 g a. i. ha^−1^ *B. eleusines* conidial agent on barnyardgrass remained between 6.38~100%, while the herbicidal activity of 0.72~45 g a. i. ha^−1^ 25 g/L penoxsulam OD stood at 2.88~100%. The result of the statistics regression (Dose–response analysis) indicated that the ED_90_ values of the *B. eleusines* conidial agent and the control agent of 25 g/L penoxsulam OD were 92.06 g a. i. ha^-1^ and 23.33 g a. i. ha^−1^, respectively. The coefficient of the specificity of the *B. eleusines* conidial agent was higher than 3.91 and closer to the herbicidal control of 2.89 and showed its high specificity between rice and barnyardgrass. It also suggested that the *B. eleusines* agent was safe for rice and highly efficient on barnyardgrass in the greenhouse.

### 2.3. Weed Control Efficacy of B. eleusines Conidial Agent with Herbicides under Field Conditions

Two-year field tests for the weed control efficacy and effect on the rice crop of *B. eleusines* conidial agent application were carried out at different experimental fields of Fuyang Experiment Station and the Shanghai Seed Breeding Center in 2011 and 2012, respectively.

There was no infection of rice plants by *B. eleusines* with eye observation and no adverse effect on the growth and development of rice plants in both experimental fields. Three weed species occurred to a substantial extent in density in the test field, i.e., barnyardgrass, 13.8 plants m^−2^, small flower umbrella sedge (*Cyperus difformis* L.), 12.6 plants m^−2^, and monochoria (*Monochoria vaginalis*), 7.0 plants m^−2^ in the Shanghai Seed Breeding Center. The occurrence of these weeds reduced rice yield (rough rice) by 15.2% in 2012 in the nontreated plot as compared with the rice yield of plots (5500.9 kg ha^−1^) where a commercial mixture of butachlor that is popular with farmers in China was treated as a control (Table 4). The application of herbicide butachlor resulted in density reduction in barnyard grass by 68.0%, small-flower umbrella sedge by 95.1%, and monochoria by 63.5%. Under such a situation of weed occurrence and the weed control efficacy of this herbicide, the spray of the *B. eleusines* conidia at a concentration of 60 g a. i. ha^−1^ showed a reduction in the density of barnyardgrass by 93.9%, small-flower umbrella sedge by 99.3%, and monochoria by 83.8% (Table 4). The application of more *B. eleusines* conidia did not significantly result in more control of barnyardgrass and small-flower umbrella sedge, the overall weed control efficacy being superior to that of the herbicide application.

The same experiment result was obtained in field lots of the Fuyang Experiment Station. We also found that there was nearly no effect on the weeds in synergist-free *B. eleusines* formulation treatment compared to treatment with a synergistic agent (Table 5). It suggested that there was a positive synergetic effect to improving *B. eleusines* conidia formulation efficiency on weeds with herbicides of penoxsulam plus bensulfuron-methyl.

## 3. Discussion

Chemical herbicides are the most commonly utilized crop-protection compounds in the world and have played an important role in effectively and quickly repressing weed populations and significantly reducing labor intensity since it was invented. However, concerns have been gradually raised about environmental risks due to the long-term heavy use of herbicides. It could cause unsustainable ecological practices including resistant weeds’ population increase and diversity decline in the rice paddy. Compared with it, bioherbicides could be more environmentally friendly and not easily lead to naturally herbicide-resistance weeds. Its disadvantage focused on the lower efficacy in the field. It might also need more time to experience fungal adsorption, colonization, germination, and growth at the host. Therefore, the weeding effect is slower than chemical herbicides. The integration of different action modes of pathogenic fungus and chemical herbicide could not only synergistically enhance the virulence of the bioherbicidal pathogens, but also significantly decrease the risk of occurring herbicide-resistance weeds. This strategy could synchronously meet the demands of government policies and consumer trends to reduce the use of synthetic pesticides and increase the use of nonchemical control methods [31]. These studies showed that a combination of *B. eleusines* with a lower rate of penoxsulam plus bensulfuron-methyl could effectively control barnyardgrass, monochoria, and small-flower umbrella sedge, and had no negative impact on the rice plant under field conditions. It suggested that this pathogen conidial agent had the potential to develop a commercial bioherbicide in the future.

The germination assay on agar provides an initial estimate of compatibility between herbicides and pathogens [32,33]. In this study, 50% quinclorac WP caused immediate losses in viability after 12 h of incubation and severe inhibition to the bioherbicide agent *B. eleusines* in the in vitro tests. However, 25 g/L penoxsulam OD and 10% bensulfuron methyl WP were not inhibitory to it. On the contrary, there was a dose-dependent increase both in the survival rate and the growth rate of *B. eleusines* in response to these two herbicides, so a tank mix of 25 g/L penoxsulam OD and 10% bensulfuron methyl WP with *B. eleusines* should be a better option to control weeds in the further greenhouse and field conditions because of the high compatibility.

The combinations of bioherbicides and synthetic herbicides can be synergistic in much research [33,34]. Some herbicides could lower the inherent defenses of the weed by inhibiting specific plant defense pathways such as pathogen-induced phytoalexin biosynthesis (phenyl-propanoid phytoalexins and steroid phytoalexins) and callose biosynthesis that stop pathogen attack, as well as many other defenses [15,35]. For example, Sharon found that a low concentration of *Alternaria cassiae* conidia elicited a hypersensitive response on the weed but not the crop after being sprayed with low levels of glyphosate together which suppressed weed phytoalexin biosynthesis [36]. Wymore et al. (1987) reported that coapplications of *C. coccodes* Wallr. and the herbicide thidiazuron to velvetleaf (*Abutilon theophrasti* Medic.) increased pathogen infection and weed control compared with either component applied alone [37]. Similarly, Peng et al. (2005) [33] also reported that coapplying propanil, quinclorac, or sethoxydim at a one-quarter rate with the pathogen at the sublethal dose of 2 × 10^7^ spores/ml achieved complete control of green foxtail, because sethoxydim, with its ability to inhibit cell division, is highly effective on young and actively growing tissues [38]. In this study, there was a positive synergetic effect to improve *B. eleusines* conidia formulation efficiency on weeds with herbicides. When *B. eleusines* conidia were sprayed at a concentration of 60 g a. i. hm^−2^ combined with a sub-rate synergist, it showed a reduction in the density of barnyardgrass by 93.9%, small-flower umbrella sedge by 99.3%, and monochoria by 83.8% (Table 4). Briefly, 25 g/L penoxsulam OD and 10% bensulfuron methyl were used as synergetic herbicides with *B. eleusines*. It seemed that penoxsulam and bensulfuron methyl could lower weed defense responses, making weeds more susceptible to pathogen attack. Therefore, a lower dose of bioherbicide agent could effectively control weeds. However, the specific synergetic mechanism needs to be further studied.

Herbicide-resistance weeds resulting from pesticides abusage in the recent several decades have caused a great threat to rice production. It could lead to no herbicide available for some superweeds. Penoxsulam is a post-emergence triazolopyrimidine sulphonamide herbicide registered for weed control in rice crops [39,40] and bensulfuron-methyl is a representative sulfonylurea herbicide that controls broadleaf weeds [41]. Both of them could inhibit acetolactate synthase (ALS) activity in susceptible plant species, and application alone easily produced herbicide resistance in agricultural practices [42,43]. The integration application of different action modes of bioherbicide agent *B. eleusines* could potentially delay the development of herbicide resistance and greatly extend the shelf life of the herbicide.

Compared to the greenhouse test, there was a decline in weed control efficiency by the *B. eleusines* agent in the field application. Schnick and Boland (2004) also found that the dandelion response was reduced after treatment with sublethal doses of 2, 4-D followed later by *P. herbarum* inoculation under field conditions [22]. The environmental conditions or other factors play a role in efficacy after combining fungal pathogens and herbicides. Therefore, some UV protective agents and others need to be added to enhance the stability and efficacy of *B. eleusines* in the field application.

The meristems of many types of grass are protected by leaf sheaths [44]. It was observed that the young leaves of barnyardgrass tend to be more resistant to *B. eleusines* despite severe injuries on lower leaves, so barnyardgrass treated with *B. eleusines* alone could not be effectively killed and often restored 14 days after treatment. However, the efficacy of weed control is significantly improved by the addition of synergetic penoxsulam and bensulfuron methyl. This is similar to reports by [33]. They found that green foxtail treated with *P. setariae* alone often recovers from initial damages. However, after being combined with sethoxydim, with its ability to inhibit cell division, they became highly effective on young and actively growing tissues [38]. These synergetic herbicide effects might be complementary to the mode of action by bioherbicide agents on weeds.

## 4. Materials and Methods

### 4.1. Fungal Inoculum

*Bipolars eleusines* was isolated from severely diseased barnyardgrass and preserved at 4 °C in the Weed Lab at the China National Rice Research Institute (CNRRI), Hangzhou, China [30]. Conidia of *B. eleusines* were produced on solid media (rice flour 4%, soybean meal 1%, Na_3_PO_4_·12H_2_O 0.2%, MgSO_4_·7H_2_O 0.1%, distilled water 40%, and 100 mL perlite) in a pallet of 45 cm (length) × 35 cm (width) × 5 cm (height) at 28 °C for 2 weeks and collected by a Mycoharvester of the Solid State Fermentation (Nanjing Institute of Agricultural Mechanization, Ministry of Agriculture and Rural Affairs, Nanjing, China) for use in further experiments.

A conidial agent of *B. eleusines* was prepared with 1 g spore powder by the addition of 0.25 mL soybean oil, sodium carboxymethylcellulose (CMC) 0.49 g, SP-20 0.1 mL, synergist A (10% bensulfuron methyl WP) 6.56 mg, and synergist B (25 g/L penoxsulam OD) 26 μL. The conidial agent of *B. eleusines* without synergists A and B was prepared as a synergist-free control.

### 4.2. Plant Preparation

Barnyardgrass seeds, harvested at maturity from the Fuyang Experiment Station of China National Rice Research Institute, Hangzhou, China, were planted onto soil where herbicides had not been used and overlaid on a 0.5~1.0 cm fine sandy soil in plastic pots (7.5 cm diameter, 6.0 cm deep). Rice seeds (Japonica Ribenqing, Indica 9311, and Glutinous rice Guixiangsinuo P106, respectively) supplied by the Seed Bank of the China National Rice Research Institute were sown after soaking and sprouting. After sowing, the soil in each pot was moistened with tap water. Then, the pots were placed in a greenhouse maintained at a temperature between 20~35 °C and humidity greater than 50%. Two days after emergence, the seedlings were thinned to 11 plants per pot and grown for foliar treatment until the rice plants reached the 3~3.5-leaf stage and the barnyardgrass reached the 2.5-leaf stage.

### 4.3. Viability of B. eleusines Conidia with Herbicide Mixtures

Commercial formulations of herbicides used in this study were detailed in Table 6. Sterile centrifuge tubes were prepared with *B. eleusines* spores and Tween-20 (2.5% *vol*/*vol*) in water. Fresh stock suspensions of the herbicides were prepared before the experiment. The maximum use rates of the herbicides were determined by the manufacturer’s label guidance and calculated assuming a 450 L ha^−1^ application volume. After a brief pre-incubation to disperse the spores, herbicide solutions were added from the stock solutions to yield the maximum labeled application rate, 1× (Table 6), or 0.5×, or 0.25×, or 0.1× (*vol*/*vol*).

Each herbicide-rate treatment was hanging dropped onto one side of the bi-concavity slide, and the herbicide-free spore suspension was dropped onto another side as the control and then put into the sterilized Petri dish (Ø 90 mm) with absorbent filter paper to incubate moistly. After 12 h of incubation, the germination of the spore was observed microscopically, and we calculated the inhibition rate of spore germination with the following formula: Inhibition rate of spore germination (%) = (1 − Nx/Ny) ∗ 100%. where Nx and Ny = data from the treated and non-treated control spore, respectively. Each treatment was replicated three times and the experiment was conducted twice.

### 4.4. Effect of Herbicide Mixtures on Mycelium Growth of B. eleusines

The chemical herbicides were diluted with sterilized water to the required concentration. A volume of 1 ml of each dilution was added into sterilized medium (2% glucose, 0.5% KNO_3_, 0.2% NaPO_3_ 12H_2_O, 0.1%MgSO_4_, 0.5% soybean powder, 1.7% agar, and water 99 mL) as herbicide media with the maximum labeled application rate of 1× (Table 6), 0.5×, 0.25×, 0.2×, or 0.1× (*vol*/*vol*), adequately mixing and putting in Petri dishes. The same volume of water was controlled. After solidifying, the mycelial discs (0.5-cm-diam agar plug) obtained from the margin of an actively growing colony of *B. eleusines* on PDA (Potato Dextrose Agar) plate culture media was inoculated on the center of the herbicide media and kept at 28 ° C under constant darkness for 14 days. Each herbicide-rate treatment was conducted with three replications. The colony diameters of *B. eleusines* were determined after 14 days of treatment. The percentage of the growth inhibition rate was calculated by comparison to herbicide-free treatment.

### 4.5. Pot Experiment in Greenhouse

To investigate the effects on rice and barnyardgrass under different dosages of the *B. eleusines* conidial agent after foliar spraying treatment, a pot experiment was conducted in the greenhouse.

The conidial agent was sprayed using a spray tower 3WPSHZ-500 (Nanjing Institute of Agricultural Mechanization, Ministry of Agriculture and Rural Affairs) at 200 kPa air pressure and a distance of about 20 cm from the plants with 0.097 m^2^ of the spray area. The *B. eleusines* conidial agent was sprayed at the rates of 3.75, 7.5, 15, 30, 60, 90, or 120 g a.i. ha^−1^ for weed control activity and 45, 90, 135, 180, 270, or 360 g a. i. ha^−1^ for rice safety evaluation, respectively. The pot was irrigated at 3 to 5 cm deep at 2 d per treatment. A commercial herbicide of penoxsulam (25 g/L a. i. by control) was applied at a rate of 0.72, 2.25, 5, 5.63, 6.75, 8.28, 22.5, or 45 g a. i. ha^−1^ for weed control activity and 2.25, 6.25, 11.25, 22.5, 45, or 67.5 g a. i. ha^−1^ for rice safety evaluation. Approximately 10 mL of conidial suspension was applied to each pot, resulting in slight runoff from the plant foliage. The control pots were sprayed with distilled water. The experiment was conducted in a completely randomized block design with four replicates for each treatment. The pots with inoculated plants were immediately covered with a plastic bag for 24 h in the greenhouse. All treatments were irrigated at 3 to 5 cm deep at 1 day per treatment.

The disease severity, the amount of viable rice and barnyardgrass, and the above-ground fresh weight of rice and barnyardgrass were recorded 4 weeks after treatment (WAT). The bioherbicideal efficacy or the inhibition rate of rice was calculated using the following formula: Reduction in fresh weight (%) = (1 − Nx/Ny) ∗ 100%, where Nx and Ny = data from treated and non-treated control plants, respectively. The coefficient of specificity was calculated using the following formula: Coefficient of Specificity = ED_10_ of crop/ED_90_ of weed.

### 4.6. Field Experiments

Field experiments were conducted at the Fuyang Experiment Station of the China National Rice Research Institute (30.079153° N, 119.934838° E), Hangzhou, in 2011 and at the Shanghai Academy of Agricultural Sciences Seed Breeding Center (31.032243° N, 121.227747° E), Shanghai, in 2012, respectively. In 2011, a conventional indica rice Yangdao 6 was cultivated as a single crop growing from May to September. In 2012, the conventional early maturing late japonica rice, “Wuyunjing 7”, was cultivated from July to December. Three-week-old rice seedlings were transplanted with 25 cm by 20 cm spacing in plots measuring 5 m by 4 m for each. The field was drained 7 days after transplanting the rice seedlings, and the rice plants were sprayed with the conidial agent of *B. eleusines*. The conidial agent combined with herbicides was sprayed at the rates of 60, 90, 120, or 180 g a. i. hm^−2^ in a spray volume of 450 L ha^−1^ water. A synergist-free conidial agent was sprayed at the rates of 90 g a. i. hm^−2^ as a control in the Fuyang Experiment Station. The field was irrigated again at 3 to 5 cm deep at 2 days after treatment. A commercial herbicide of butachlor (60% a. i. by weight) that is popular with farmers for weed control in rice fields in Zhejiang and Shanghai, China was applied at a rate of 990 g a. i. of butachlor per ha, 5 days in Shanghai and 7 days in CNRRI after transplanting. The non-treated control was sprayed with water. Manual weeding was carried out 15 days after application. The experimental units were arranged in a randomized block design with three replications. The density and aboveground fresh weights of weeds per 0.25 m^2^ were recorded at 4 WAT. The weed control efficacy, i.e., the reduction in density or weight of the weed, of the treatments was calculated as described previously. The whole plot was manually harvested, and the rice grain yield was recorded.

### 4.7. Data Analysis

All data analysis was performed using an SPSS 13.0 statistical package. The statistical comparisons of data for the percentage of the inhibition rate or reduction rate in Table 1, Table 2, Table 4, and Table 5 were performed via analysis of variance, and the data in Table 3 through regression analysis. The treatment means in Table 1 and Table 2, after uniformly normalizing to positive decimals, then square root, and arcsin transformation to normalize the variance, were compared by Tukey’s multiple range tests at a 5% level of significance. The treatment means of the percentage of the reduction rate in Table 4 and Table 5 were compared by Duncan’s multiple range tests at a 5% level of significance after the same conversion. The results were back-transformed to the original ratings for presentation. The statistical analyses for the rice yield in Table 4 and Table 5 were performed without transformation.

## 5. Conclusions

In brief, 25 g/L penoxsulam OD plus 10% bensulfuron-Methyl WP were compatible with *B. eleusines* in an in vitro test. The conidial agent of *B. eleusines*, by the addition of 25 g/L penoxsulam OD plus 10% bensulfuron-Methyl WP, was safe on three rice varieties, including japonica, indica, and glutinous rice, and showed high specificity between rice and barnyardgrass under greenhouse conditions. Field experiments showed that *B. eleusines* combined with chemical herbicides could improve the biological efficacy of the *B. eleusines* conidial agent. These results were critical for the commercial application of the *B. eleusines* conidial agent in the future.

## Figures and Tables

**Table 1 plants-11-02659-t001:** Inhibition rate of survival and radial growth of *B. eleusines* by herbicides.

**Product**	**Mixing Rate**	**Inhibition Rate of Conidial Germination ± SD (%)**	**Interacts**	**Inhibition Rate of Mycelium Radial Growth ± SD (%)**	**Interacts**
Quinclorac 50% WP	0.2×	0.7 ± 1.6 ^b &^	NO *	−7.1 ± 1.3 ^b^	NO
	0.5×	36.8 ± 7.2 ^a^	- **	16.0 ± 2.3 ^a^	-
	1× ^@^	36.7 ± 4.1 ^a^	-	17.1 ± 6.0 ^a^	-
Cyhalofop-butyl 100 g/L EC	0.2× 0.5×	−5.0 ± 2.4 ^a^ −19.6 ± 1.6 ^b^	NO + ***	7.1 ±± 2.7 ^b^ 38.1 ± 1.4 ^a^	NO -
	1×	−20.0 ± 0.5 ^b^	+	38.6 ± 6.7 ^a^	-
Penoxsulam 25 g/L OD	0.2× 0.5×	−11.0 ± 0.4 ^b^ −18.7 ± 0.6 ^d^	+ +	−6.4 ± 6.8 ^b^ 24.3 ± 2.2 ^a^	NO -
	1×	−16.5±0.7 ^c^	+	16.4±4.0 ^ab^	-
Bensulfuron methyl 10% WP	0.2× 0.5×	1.5 ± 1.3 ^b^ 23.8 ± 1.3 ^a^	NO -	−29.3 ± 1.7 ^a^ −41.9 ± 4.5 ^a^	+ +
	1×	23.5±3.3 ^a^	-	−65.0±6.1 ^b^	+
Pyrazosulfuron-ethyl 10% WP	0.2× 0.5×	−6.4 ± 1.8 ^b^ 35.5 ± 0.9 ^a^	NO -	7.4 ± 2.0 ^a^ −1.2 ± 1.3 ^ab^	NO NO
	1×	34.6±0.8 ^a^	-	−4.0±2.4 ^b^	+
Quinclorac 50% WP + Bensulfuron methyl 10% WP	0.1×+0.1×	7.9 ± 3.8 ^c^	-	−9.8 ± 2.8 ^a^	NO
	0.25×+0.25×	18.6 ± 2.6 ^b^	-	−30.0 ± 8.4 ^a^	NO
	0.5× +0.5×	48.4 ± 3.1 ^a^	-	−21.2 ± 4.2 ^a^	NO
Cyhalofop-butyl 100 g/L EC + Bensulfuron methyl 10% WP	0.1× +0.1×	10.2 ± 2.9 ^a^	NO	−22.9 ± 3.1 ^b^	NO
	0.25× +0.25×	1.1 ± 3.8 ^a^	NO	21.7 ± 4.8 ^a^	-
	0.5× +0.5×	0.5 ± 1.4 ^a^	NO	21.9 ± 4.2 ^a^	-
Penoxsulam 25 g/L OD + Bensulfuron methyl 10% WP	0.1× +0.1×	1.0 ± 3.3 ^a^	NO	−10.5 ± 3.6 ^a^	+
	0.25× +0.25×	0.7 ± 3.8 ^a^	NO	−42.4 ± 4.9 ^a^	+
	0.5× +0.5×	−12.2 ± 1.4 ^b^	+	−111.0 ± 5.9 ^b^	+
Quinclorac 50% WP + Pyrazosulfuron-ethyl 10% WP	0.1× +0.1×	39.0 ± 3.6 ^a^	-	7.6 ± 4.4 ^a^	NO
	0.25× +0.25×	11.5 ± 3.8 ^b^	-	4.0 ± 3.8 ^a^	NO
	0.5× +0.5×	40.6 ± 4.8 ^a^	-	3.8 ± 2.5 ^a^	NO
Cyhalofop-butyl 100 g/L EC + Pyrazosulfuron-ethyl 10% WP	0.1× +0.1×	12.4 ± 2.9 ^a^	NO	10.0 ± 5.5 ^b^	-
	0.25× +0.25×	0.7 ± 5.3 ^a^	NO	47.6 ± 3.8 ^a^	-
	0.5× +0.5×	−2.7 ± 4.0 ^a^	NO	56.0 ± 9.2 ^a^	-
Penoxsulam 25 g/L OD + Pyrazosulfuron-ethyl 10% WP	0.1× +0.1×	−9.1 ± 0.8 ^b^	+	−0.7 ± 4.6 ^a^	NO
	0.25× +0.25×	−4.5 ± 0.6 ^ab^	NO	−0.2 ± 4.0 ^a^	NO
	0.5× +0.5×	−6.5 ± 0.4 ^ab^	NO	−8.1 ± 4.3 ^a^	NO
CK (0.05% Tween-20)		80.1 ± 2.1 ^#^		4.20 ± 0.20 ^##^	

^@^ 1× refers to the maximum application rate, assuming a 300 L ha^−1^ application volume. Briefly, 0.1×, 0.2×, 0.25×, and 0.5× indicate one-tenth, one-fifth, one-quarter, one half the maximum rate, respectively. ^&^ Means within a column of the same row followed by the same superscript letter are not significantly different at 0.05. * NO refers to no significant effects. **− refers to inhibition effects. *** + refers to synergetic effects. # and ## values refer to the conidial germination and mycelium radial growth of *B. eleusines* in non-treated control of 0.05% Tween-20, respectively.

**Table 2 plants-11-02659-t002:** Inhibition rate of fresh weight of *B. eleusines* conidial agent and 25 g/L penoxsulam OD on three rice varieties.

Agent	Dosage g a.i. ha ^−1^	Inhibition Rate of Plant Fresh Weight ± SD (%)		
		Japonica Rice (Ribenqing)	Glutinous Rice (Guixiangsinuo P106)	Indica Rice (9311)
Conidial agent of *B. eleusines*	45	−3.95 ± 1.24	−2.29 ± 0.78	0.24 ± 1.39
	90	−2.38 ± 0.89	−2.26 ± 1.36	−4.96 ± 2.36
	135	−2.31 ± 1.33	−4.96 ± 1.78	−5.15 ± 1.58
	180	−4.57 ± 1.84	−5.35 ± 1.46	−2.24 ± 1.02
	270	−5.35 ± 1.54	−6.02 ± 3.21	−0.66 ± 2.33
	360	0.38 ± 2.12	−3.40 ± 1.25	−4.15 ± 2.14
Penoxsulam 25 g/L OD	2.25	−3.19 ± 1.36	3.15 ± 3.69	−1.49 ± 1.36
	6.75	−3.90 ± 2.67	−2.25 ± 2.21	−3.42 ± 2.66
	11.25	−3.15 ± 1.78	−0.07 ± 1.39	−3.67 ± 3.15
	22.5	−6.16 ± 3.11	−1.71 ± 0.58	−2.68 ± 1.04
	45	−2.37 ± 1.83	0.28 ± 1.06	−2.53 ± 2.05
	67.5	1.64 ± 1.45	2.10 ± 1.10	−5.66 ± 3.02

**Table 3 plants-11-02659-t003:** Determination of ED_90_ value of *B. eleusines* conidial agent and 25 g/L penoxsulam OD on barnyardgrass.

Agent	Dose g a. i. ha^−1^	Inhibition Rate of Fresh Weight %	Regression Equations	Correlation Coefficient R	ED_90_Value g a. i. ha^−1^
Conidial agent of *B. eleusines*	3.75	6.38	Y = −18.99 + 24.10 lnX	0.9733	92.06
	7.5	25.15			
	15	59.85			
	30	67.99			
	60	73.53			
	90	94.27			
	120	96.29			
	180	100.00			
Penoxsulam 25 g/L OD	0.72	2.88	Y = 9.08 + 25.69 lnX	0.9690	23.33
	2.25	24.18			
	5	38.85			
	5.63	62.59			
	6.75	64.18			
	8.28	61.64			
	22.5	97.64			
	45	100.00			

**Table 4 plants-11-02659-t004:** Effects of *B. eleusines* agent with herbicides on weed control and rice yield at Shanghai Seed Breeding Center in 2012.

**Treatment**		**Density Reduction Rate ± SD (%)**				**Fresh Weight Reduction Rate ± SD (%)**				**Rice Yield**
	**Dosage**	**Barnyardgrass**	**Monochoria**	**Small-Flower Umbrella Sedge**	**Total Weeds**	**Barnyardgrass**	**Monochoria**	**Small-Flower Umbrella Sedge**	**Total Weeds**	**(kg ha ^-1^)**
*B. eleusines* conidial agent	60	93.9 * ± 3.5 ^a^ **	83.8 ± 1.4 ^b^	99.3 ± 0.7 ^a^	92.6 ± 1.0 ^b^	96.7 ± 2.0 ^a^	86.4 ± 1.2 ^c^	99.8 ± 0.2 ^a^	96.0 ± 0.5 ^a^	5746.4 ± 84.0 ^a^
	90	95.0 ± 3.2 ^a^	86.7 ± 1.5 ^ab^	96.9 ± 1.8 ^a^	92.9 ± 1.3 ^ab^	96.8 ± 2.3 ^a^	90.6 ± 1.1 ^bc^	98.3 ± 1.1 ^a^	96.1 ± 0.8 ^a^	5750.7 ± 57.4 ^a^
	120	97.3 ± 2.8 ^a^	88.4 ± 2.1 ^ab^	96.8 ± 3.2 ^a^	94.0 ± 2.3 ^ab^	98.2 ± 1.8 ^a^	89.8 ± 1.3 ^bc^	98.4 ± 1.6 ^a^	96.5 ± 1.3 ^a^	5749.5 ± 61.7 ^a^
	180	100.0 ± 0.0 ^a^	91.7 ± 2.4 ^a^	100.0 ± 0.0 ^a^	97.2 ± 0.8 ^a^	100.0 ± 0.0 ^a^	93.5 ± 0.6 ^ab^	100.0 ± 0.0 ^a^	98.6 ± 0.1 ^a^	5803.4 ± 70.4 ^a^
Butachlor	990	68.0 ± 4.2 ^b^	63.5 ± 4.9 ^c^	95.1 ± 2.0 ^a^	77.3 ± 2.0 ^c^	73.1 ± 3.8 ^a^	68.1 ± 2.8 ^d^	97.3 ± 0.9 ^a^	83.1 ± 2.0 ^b^	5500.9 ± 50.3 ^b^
Manual weeding	-	96.7 ± 1.4 ^a^	93.4 ± 0.7 ^a^	97.2 ± 1.7 ^a^	95.8 ± 0.9 ^ab^	98.3 ± 0.7 ^a^	95.3 ± 0.3 ^a^	98.9 ± 0.7 ^a^	97.9 ± 0.6 ^a^	5823.5 ± 35.5 ^a^
Non-treated control	Water	0 (45.3 ***) ^c^	0 (60.3) ^d^	0 (71.0) ^b^	0 (176.5) ^d^	0 (139.1) ^c^	0 (89.9) ^e^	0 (193.7) ^b^	0 (422.8) ^c^	4776.9 ± 80.0 ^c^

* Values present the means of three replicates; ** within a column values with the same superscript letter are not significantly different at 0.05 level; *** Values in the brackets indicate the means of three replicates of weed density plants per 0.75 m^2^ and the fresh weight of aboveground plants per 0.75 m^2^ in non-treated control plots data were recorded 4 weeks after treatment.

**Table 5 plants-11-02659-t005:** Effects of *B. eleusines* agent with herbicides on weed control and rice yield at the Fuyang Experiment Station in 2011.

Treatment		Density Reduction Rate ± SD (%)			Fresh Weight Reduction Rate ± SD (%)			Rice Yield
	Dosage	Barnyardgrass	Monochoria	Small-Flower Umbrella Sedge	Barnyardgrass	Monochoria	Small-Flower Umbrella Sedge	kg ha ^−1^
*B. eleusines* conidial agent	60	87.5 ± 6.4 * ^a^ **	80.4 ± 9.2 ^a^	100.0 ± 0.0 ^a^	85.7 ± 7.3 ^a^	71.8 ± 4.2 ^a^	100.0 ± 0.0 ^a^	8987.1 ± 308.3 ^a^
	90	88.3 ± 5.9 ^a^	65.8 ± 8.6 ^a^	100.0 ± 0.0 ^a^	89.8 ± 6.1 ^a^	72.3 ± 6.9 ^a^	100.0 ± 0.0 ^a^	8734.8 ± 194.1 ^a^
	120	89.1 ± 2.1 ^a^	84.8 ± 6.6 ^a^	98.2 ± 1.8 ^a^	86.5 ± 2.8 ^a^	86.4 ± 7.3 ^a^	96.4 ± 3.6 ^a^	8671.0 ± 66.4 ^a^
	180	100.0 ± 0.0 ^a^	87.3 ± 0.6 ^a^	94.6 ± 5.4 ^a^	100.0 ± 0.0 ^a^	93.0 ± 0.5 ^a^	95.5 ± 4.5 ^a^	8737.5 ± 149.1 ^a^
*B. eleusines* synergist-free conidial agent	90	16.4 ± 12.2 ^c^	7.6 ± 8.4 ^c^	49.1 ± 15.5 ^b^	13.1 ± 11.6 ^bc^	(−136.5) ± 27.6 ^d^	21.7 ± 12.8 ^bc^	7398.7 ± 51.7 ^c^
Butachlor	990	46.1 ± 11.1 ^b^	48.7 ± 11.4 ^b^	40.1 ± 13.9 ^b^	32.2 ± 23.4 ^b^	12.0 ± 6.1 ^bc^	43.6 ± 20.3 ^b^	7919.3 ± 61.0 ^b^
Manual weeding	-	100.0 ± 0.0 ^a^	81.0 ± 5.0 ^a^	100.0 ± 0.0 ^a^	100.0 ± 0.0 ^a^	53.7 ± 20.0 ^ab^	100.0 ± 0.0 ^a^	9025.7 ± 87.8 ^a^
Non-treated control	Water	0 (14.22 ***) ^c^	0 (17.56) ^c^	0 (18.56) ^c^	0 (45.75) ^c^	0 (48.42) ^c^	0 (9.67) ^c^	7032.0 ± 132.4 ^c^

* Values present the means of three replicates; ** within a column values with the same superscript letter are not significantly different at 0.05 level; *** Values in the bracket indicate the means of three replicates of weed density plants per 0.75 m^2^ and the fresh weight of aboveground plants per 0.75 m^2^ in non-treated control plots data were recorded 4 weeks after treatment.

**Table 6 plants-11-02659-t006:** Commercial formulations of herbicides and adjuvants evaluated for compatibility with *B. eleusines*.

Herbicide Product	Supplier	Active Ingredient	Classification/Mode of Action	Maximum Use Rate g a.i.ha^−1^	Application Rate
Quinclorac 50% WP	Zhejiang Tianyi Agrochemical Co., Ltd., Shaoxing, China	quinclorac	quinolinecarboxylic acid/hormone type	375	0.1×, 0.2×, 0.25×, 0.5×, 1× ^a^
Cyhalofop-butyl 100 g/L EC	Dow AgroSciences, Indianapolis, IN, USA	cyhalofop-butyl	aryloxy phenoxy propionate, APP/ACCase	105	0.1×, 0.2×, 0.25×, 0.5×, 1× ^b^
Penoxsulam 25 g/L OD	Dow AgroSciences, Indianapolis, IN, USA	penoxsulam	triazolopyrimidine sulfonamides/ALS	30	0.1×, 0.2×, 0.25×, 0.5×, 1× ^b^
Bensulfuron methyl 10% WP	Zhejiang Tianyi Agrochemical Co., Ltd., Shaoxing, China	bensulfuron methyl	sulfonylureas/ALS	30	0.1×, 0.2×, 0.25×, 0.5×, 1× ^a^
Pyrazosulfuron-ethyl 10% WP	Nissan Chemical Corporation, Japan	pyrazosulfuron-ethyl	sulfonylureas/ALS	30	0.1×, 0.2×, 0.25×, 0.5×, 1× ^a^
Tween-20	Sigma Aldrich Co.	polysorbate 20	-		0.025 ^b^

a. Mass: volume; b. Volume: volume.

## Data Availability

Not applicable.

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
