# Peer review of "Improved Bioherbicidal Efficacy of Bipolaris eleusines through Herbicide Addition on Weed Control in Paddy Rice"

_plants, 2022, doi:10.3390/plants11192659_

Round 1
Reviewer 1 Report
The page numbering is all wrong. You have 2 pages labeled “2 of 15” and other similar problems. The line numbering is all wrong also – sometimes it is continuous from one page to another, then it will randomly start over at 1.
Line 33: Why do you say “environmental” pollution? Is there some kind of pollution that is NOT environmental?
Line 33: I can’t really understand how replacing (potentially) a herbicide with a bioherbicide will help with the loss of biodiversity. This idea isn’t really the point of your manuscript anyway, so it’s just a distraction.
Line 43: Isn’t specificity a good thing? Why are you calling it a problem here?
Line 52: Sencor is a brand name. You should probably stick to the generic name or the active ingredient to be consistent with the rest of the paragraph.
Line 78: “Separate”
Table 1: Some of these results are puzzling. I can understand that some herbicides are very well-tolerated by the fungus, but some of these dramatic POSITIVE effects of the herbicide on the fungus are hard to understand.
Line 95: Why are you only at 2.5 to 3.5 leaf stage if this is 4 weeks after application.
Table 2: Should be “Japonica Rice” in the heading, not Rcie.
Table 2: I can’t tell what these values are – plant weight? Yield?
Line 91-94: This part doesn’t make sense. It looks here like you are adding two herbicides as “synergists”. These herbicides may be synergists, but some of the experiments must have used the fungus without the additional herbicides, right?
Line 97: I have no idea what “planted on soil useless of herbicides” means
Line 28: The control plants being sprayed with water should have been sprayed with the surfactants that were used elsewhere.
Table 5: “Yield” instead of “Yiled”.
Table 5: I would probably just report the yield instead of the % increase of yield. If you are going to report the % increase, you should state what the control yield is.
Table 5. Instead of saying “the same as Table 4”, just write what it means.
Table 5. Change the vertical spacing on this table – it is hard to read across the line
Reviewer 2 Report
Review (Manuscript ID: plants-1927127)
The paper presents quite interesting results and the research concerns innovative solutions in weed control however the manuscript is not ready for publication. I recommend major revision of the manuscript.
The title is adequate to the topic of research. In the Introduction more information about Bipolaris eleusines from the scientific literature should be given. The objective of the study wasn’t formed clearly, should be improved. Please provide the hypothesis tested. The sections “Materials and methods” and “Results” should be carefully corrected. They are described imprecisely, some information is missing. References should be actualized (about one third of literature positions are older than 20 years). English should be checked. More specific comments are given below.
Specific comments:
lines 52 (page 2, Introduction), 94 (page 2, Material and methods), Table 6 and other places in the manuscript: Spelling of herbicide names should be unified - herbicides names should be capitalised and active ingredients names should be lowercase.
Table 1, 5 – please provide the standard error values
Table 2 – please provide the unit for the inhibition rate
lines 119-120 - It should be the title of the subsection 2.3
line 62, page 2, Discussion – herbicide resistance not drug resistance
subsection 4.6 Field Experiments – what were the wheather conditions during the field experiment? It’s a very important information for mycoherbicides. Were there plots without B. eleusines and/or herbicide spraying in the field experiment? I haven’t noticed them in the methods description, however in the Table 4 and 5 the results for non-treated check are given.
subsection 4.7. Data Analysis – was the distribution normal for all traits? With which test was the normality of distribution checked (if any)? Why the Duncan test was used and not Tuckey – Duncan is a quite liberal test and some differences between mean can be overestimated. Majority of the results are expressed in % (inhibition rate of conidial germination, inhibition rate of mycelium radial growth, density reduction rate, fresh weight reduction rate etc.) - were these data transformed before the statistical analysis?

Reviewer 3 Report
The authors investigated and discussed “Improved bioherbicidal efficacy of Bipolaris eleusines on barnyardgrass (Echinochloa crusgalli) by addition of herbicide”. The authors mentioned well the matter of the research showing the importance of Bipolaris eleusines as a biocide. This seems of a great importance but many changes/clarifications and English revision are needed before publishing.
Title: The authors mentioned in the title only barnyardgrass while their experiments contained Monochoria vaginalis and Cyperus difformis L. too. Please, rewrite the title in accordance with the other species
Abstract:
Line 13: What is OF?
Line 17: “high” not in italic
Keywords: Please, rewrite all of them in lowercase
Introduction:
Line 39: The statement “supplement some of the current chemical herbicides…” is not clear
Line 48: For the first time in the text is required to mention the full name of 2,4 D and d even if they are known
Line 50: The full name of “MCPA”
Results
Line 81-82: Are the results “50.6%-82 61.3% inhibition” reported in the table 1?
Materials and methods
Line 94: OF?
Line 101: Please, change “seeding” to “sowing”
Line 102: “seeded dishes”, please, check the phrase “seeded” and then “dishes” should be “pots”?
Line 108: Add “s” to “spore”
Line 117: How the spore germination rate was calculated?
Page 5-Line 9: What is PDA?
Page 5-Line 23+31: “at 2 d treatment” add 2per” before treatment
Page 5-Line 26: “Rice safety evaluation”, How the rice safety was evaluated?
Page 5-Line 28: “Control pots” instead of “check pots”
Page 5-Line 39-41: Please, add the coordinates (latitude/longitude) of the location.
Page 5-Line 49: “as control” instead of “as check”
Page 5-Line 49: Which substance was used as a synergist–free conidia agent
Conclusions
Please, add the conclusions
Round 2
Reviewer 1 Report
The author's response to my review was very helpful. I'm satisfied with the corrections.
Author Response
I am deeply appreciated for your review again.
Reviewer 2 Report
The manuscript has been improved by the authors. I recommend minor revision because the names of the herbicide product should be corrected. If the authors write about commercially available herbicide (product) they should use capital letter at the beggining of the name of the product, f. ex. Quinclorac 50% WP but if they write about active ingredient, they should write quinclorac (no capital letter at the beggining). The same rule concerns other herbicide products or active ingredients described in the manuscript.
Author Response
Answer to Reviewer’s Comments and Suggestions
Q1. The manuscript has been improved by the authors. I recommend minor revision because the names of the herbicide product should be corrected. If the authors write about commercially available herbicide (product) they should use capital letter at the beggining of the name of the product, f. ex. Quinclorac 50% WP but if they write about active ingredient, they should write quinclorac (no capital letter at the beggining). The same rule concerns other herbicide products or active ingredients described in the manuscript.
A1. Thank you for your advice. They all have been revised in the manuscript as below.
Table 6. Commercial formulations of herbicides and adjuvants evaluated for compatibility with B. eleusines.
|
Herbicide product |
Supplier |
Active ingredient |
Classification/mode of action |
Maximum use rate g a.i.ha-1 |
Application rate |
|
Quinclorac 50% WP
|
Zhejiang Tianyi Agrochemical Co., Ltd., Shaoxing, China
|
quinclorac |
quinolinecarboxylic acid/ hormone type |
375 |
0.1×, 0.2×, 0.25×, 0.5×, 1×a |
|
Cyhalofop-butyl 100g/L EC
|
Dow AgroSciences, Indianapolis, IN, USA |
cyhalofop-butyl |
aryloxy phenoxy propionate,APP/ ACCase |
105 |
0.1×, 0.2×, 0.25×, 0.5×, 1×b |
|
Penoxsulam 25g/L OD |
Dow AgroSciences, Indianapolis, IN, USA |
penoxsulam |
triazolopyrimidine sulfonamides /ALS
|
30 |
0.1×, 0.2×, 0.25×, 0.5×, 1×b |
|
Bensulfuron methyl 10% WP |
Zhejiang Tianyi Agrochemical Co., Ltd., Shaoxing, China
|
bensulfuron methyl |
sulfonylureas /ALS |
30 |
0.1×, 0.2×, 0.25×, 0.5×, 1×a |
|
Pyrazosulfuron-ethyl 10% WP |
Nissan Chemical Corporation, Japan
|
pyrazosulfuron-ethyl |
sulfonylureas/ALS |
30 |
0.1×, 0.2×, 0.25×, 0.5×, 1×a |
|
Tween-20 |
Sigma Aldrich Co. |
polysorbate 20 |
- |
|
0.025b |
- Mass:volume; b. Volume:volume.

Reviewer 3 Report
A few changes and modifications are required before publishing.
Title: I suggest the following title for the manuscript "Improved bioherbicidal efficacy of Bipolaris eleusines through herbicide addition on weed control in paddy rice "
Page 5-Line 23+31: Please, add "per" before treatment in “at 2 d treatment”. Then, the phrase will be "at 2 d per treatment". Sorry, in my comment, I wrongly typed 2per” instead of "per".
4.6. Field Experiments: Replace “check” with “control” in “Non-treated check was sprayed with water”
Author Response
Answer to Reviewer’s Comments and Suggestions
A few changes and modifications are required before publishing.
Q1. Title: I suggest the following title for the manuscript "Improved bioherbicidal efficacy of Bipolaris eleusines through herbicide addition on weed control in paddy rice "
A1. Thank you for your advice. The title has been revised as “Improved bioherbicidal efficacy of Bipolaris eleusines through herbicide addition on weed control in paddy rice”. It is exactly more appropriate.
Q2. Page 5-Line 23+31: Please, add "per" before treatment in “at 2 d treatment”. Then, the phrase will be "at 2 d per treatment". Sorry, in my comment, I wrongly typed 2per” instead of "per".
A2. It has been revised in the manuscript as below.
The pot was irrigated at 3 to 5 cm deep at 2 d per treatment. A commercial herbicide of penoxsulam (2.5% a. i. by control) was applied at a rate of 0.72、2.25、5、5.63、6.75、8.28、22.5 or 45 g a. i. ha-1 for weed control activity and 2.25、6.25、11.25、22.5、45 or 67.5 g a. i. ha-1 for rice safety evaluation. Approximately 10 ml of conidial suspension were applied to each pot, resulting in slight run off from the plant foliage. Control pots were sprayed with distilled water. The experiment was conducted in a completely randomized block design with four replicates for each treatment. Pots with inoculated plants were immediately covered with a plastic bag for 24 h in the greenhouse. All treatments were irrigated at 3 to 5 cm deep at 1 d per treatment.
Q3. 4.6. Field Experiments: Replace “check” with “control” in “Non-treated check was sprayed with water”
A3. It has been revised as “Non-treated control was sprayed with water” from nontreated check.
Non-treated control was sprayed with water.
